

# Global joint assimilation of GRACE and SMOS for improved estimation of root-zone soil moisture and vegetation response

Siyuan Tian[1,2], Luigi J. Renzullo[2], Albert I.J.M. van Dijk[2], Paul Tregoning[1], and Jeffrey P. Walker[3]

[1]Research School of Earth Science, Australian National University, Acton, ACT, 2601
[2]Fenner School of Environment and Society, Australian National University, Acton, ACT, 2601
[3]Department of Civil Engineering, Monash University, Clayton, VIC, 3800

**Correspondence:** Siyuan Tian (siyuan.tian@anu.edu.au)

**Abstract.** The lack of direct measurement of root-zone soil moisture poses a challenge to the large-scale prediction of ecosystem response to variation in soil water. Microwave remote sensing capability is limited to measuring moisture content in the uppermost few centimetres of soil. In contrast, GRACE (Gravity Recovery and Climate Experiment) mission detected the variability in storage within the total water column, which is often dominated by groundwater variation. However, not all vegetation
communities can access groundwater. In this study, satellite-derived water content from GRACE and SMOS were jointly assimilated into an ecohydrological model to better predict the impact of changes in root-zone soil moisture on vegetation vigour. Overall, the accuracy of root-zone soil moisture prediction though the joint assimilation of surface soil moisture and total water storage retrievals showed improved consistency with ground-based soil moisture measurements and satellite-observed greenness when compared to open-loop estimates (i.e. without assimilation). For example, the correlation between modelled and
in-situ measurements of root-zone moisture increased by 0.1 on average over grasslands and croplands. Improved correlations were found between vegetation greenness and soil water storage derived from the joint assimilation with an increase up to 0.47 over grassland compared to open-loop estimates. Joint assimilation results show a more severe deficit in soil water in eastern Australia, western North America and eastern Brazil over the period of 2010 to 2015 than the open-loop, consistent with the satellite-observed vegetation greenness. The assimilation of satellite-observed water content contributes to more accurate
knowledge of soil water availability, providing new insights for monitoring hidden water stress and vegetation response.

## 1  Introduction

Water is a growth-limiting resource that impacts over 40% of Earth's vegetated surface (Nemani et al., 2003). Vegetation productivity and water stress are strongly coupled by the interactions between soil moisture, photosynthesis, transpiration, interception and hydraulic redistribution (Porporato et al., 2004). The amount of water available to support plant growth and
buffer against rainfall deficiencies largely determines the length of the growing period (Leenaars et al., 2018). Although some vegetation species have roots that can grow to tens of meters depth (Canadell et al., 1996), most plants have roots that are contained in the upper 2m of the soil column, and thus cannot access the deeper water stores (Tokumoto et al., 2014). For



example, Dunne and Willmott (1996) derived a global distribution map of plant-extractable soil water capacity based on soil-water retention properties, soil texture and organic content estimates and found that less than 150 mm of the water capacity can be accessed by the plants over 90% of the vegetated area. Wang-Erlandsson et al. (2016) estimated root-zone storage capacity with satellite-based evaporation data and found large root-zone storage capacity over semi-arid regions. The duration of water

stress and the vertical distribution of soil moisture determine the vegetation vigour to a large extent in drylands (Canadell et al., 1996). Stress due to limited plant-available water can trigger a reduction in photosynthesis, which in turn leads to reduced productivity and increased vegetation mortality. The increasing deficit in deep soil water under a changing climate may further intensify ecological droughts during the growing season and may cause shifts in vegetation and ecosystem service delivery (Schlaepfer et al., 2017). There is a compelling need to quantify the vegetation responses to water scarcity for improved

assessment of climate change impacts at large scales (Breshears et al., 2005).

Wang et al. (2007) and Santos et al. (2014) investigated different responses of vegetation vigor to ground-based root-zone soil moisture observations at different depths. The absence of widespread direct observations of root-zone soil moisture has limited studies on the impacts of soil water availability on the functions in terrestrial ecosystems at regional to global scale. Soil moisture simulations and satellite water content observations from the uppermost soil layer to the total water column

have been used to quantify the water driven surface vegetation greenness variability (Chen et al., 2014; Yang et al., 2014; Andela et al., 2013; Laio et al., 2001; Xie et al., 2016; Wang et al., 2007). However, model-simulated soil moisture profile estimates are highly uncertain due to the necessary simplification of processes and parameterization (Porporato et al., 2004). Soil moisture observations from in-situ monitoring networks or satellite observations are generally spatially, vertically and temporally constrained by the instruments. Satellite soil moisture retrievals from microwave sensors such as SMOS (Soil

Moisture and Ocean Salinity) only provide the soil moisture in the uppermost soil layer and are limited by the errors introduced by soil type, canopy cover and surface roughness (Narayan et al., 2004; Houser et al., 1998). In contrast, the GRACE (Gravity Recovery and Climate Experiment) mission provided integrated water storage change including water above and under the surface through mapping anomalies in the changing Earth's gravity field (Tapley et al., 2004). It has been observed that there is almost no time lag between surface greenness and GRACE-observed total water storage change over majority of eastern

Australia (Yang et al., 2014). Conversely, Chen et al. (2014) found that vegetation greenness typically lags soil moisture at less than 10 cm depth by one month over mainland Australia using merged satellite soil moisture products (Liu et al., 2012). This discrepancy in the time lags indicates that vegetation responds differently to variations in surface soil moisture and total water storage. The quantification of vegetation response to soil water availability at large scale therefore remains challenging without accurate soil moisture profile estimations.

Observations of near-surface soil moisture have been successfully integrated into land surface models to correct modelled soil moisture deficiencies using varying assimilation techniques (Walker and Houser, 2001; Crow et al., 2008; Dumedah et al., 2015; Sabater et al., 2007; Renzullo et al., 2014). Active/radar and passive/radiometer observations were jointly assimilated to improve surface soil moisture and root-zone soil moisture with optimal accuracy and spatial coverage by Draper et al. (2012) and Lievens et al. (2017). Significant improvements were mainly found for shallow root-zone estimation at 0-30 cm



(Draper et al., 2012; Renzullo et al., 2014), with less benefit for deeper soil layers. Conversely, GRACE-observed total water storage anomalies were successfully assimilated or otherwise combined with model simulations for improved deep soil and groundwater estimation (Zaitchik et al., 2008; Khaki et al., 2017; Schumacher et al., 2018; Tangdamrongsub et al., 2015; Girotto et al., 2017; van Dijk et al., 2014; Tangdamrongsub et al., 2018), but with typically marginal improvements for surface

and shallow soil moisture (Tangdamrongsub et al., 2018; Li et al., 2012; Girotto et al., 2017; Tian et al., 2017). Recently, near-surface soil moisture and total water storage observations were jointly assimilated into a water balance model over Australia and demonstrated consistently improved water storage profile estimates, especially in the root-zone soil moisture estimates (Tian et al., 2017).

In this study, satellite-observed soil moisture and changes in total water storage were jointly assimilated into a global ecohydro-

10 logical model following the approach of Tian et al. (2017) and extended with several further innovations. We investigated the impacts of assimilating satellite water content retrievals on the estimation of surface and root-zone soil moisture and evaluated with ground-based soil moisture measurements. The relationship between vegetation vigor and soil water availability was assessed with satellite-observed greenness and observation-constrained root-zone soil moisture estimates for different vegetation types. The performance of the joint assimilation is compared against the open-loop model and alternative assimilation methods.

Satellite-observed greenness is used to quantify spatially different responses of vegetation to soil water availability at different depths, thereby determining the soil layer best reflecting the surface greenness. The interannual patterns of root-zone soil water storage are compared with the trends in vegetation greenness to investigate the potential of using accurate soil water storage estimates for explaining and anticipating vegetation greenness and productivity.

## 2 Materials

### 2.1 Ecohydrological model

The World-Wide Water (W3) model (van Dijk et al., 2013b) (available at http://wald.anu.science) is a one-dimensional, grid-based distributed ecohydrological model that simulates water balance and water-related vegetation dynamics. It was adapted from the operational Australian Water Resources Assessment Landscape (AWRA-L) model (van Dijk, 2010) used for water resource information services. A $0.25° \times 0.25°$ global gridded Multi-Source Weighted-Ensemble Precipitation (MSWEP) data

set derived by merging gauge, satellite and reanalysis data (Beck et al., 2017) was used as the only water input in the system. The $0.5° \times 0.5°$ WFDEI (WATCH Forcing Data methodology applied to ERA-Interim) meteorological forcing data set (Weedon et al., 2014) includes radiation, air temperature, wind speed, and surface pressure and these were resampled to be consistent with the resolution of precipitation. Soil and vegetation water and energy fluxes were simulated separately for deep-rooted and shallow-rooted vegetation to consider different rooting and water uptake behavior. The soil water store was partitioned



into three layers, namely, top, shallow and deep soil to describe the plant available water, approximately 0–5cm, 0.05–1m, and 1–10m in depth respectively. The unconfined groundwater and surface water stores were simulated at grid cell level.

## 2.2 Land cover types

The 2010 land cover types of each pixel were characterized by the MODIS (Moderate Resolution Imaging Spectroradiometer)
global IGBP (International Geosphere–Biosphere Programme) land cover classifications (MCD12Q1) at $5' \times 5'$ resolution (Channan et al., 2014). The number of pixels at $5' \times 5'$ resolution for each land cover type in the entire corresponding $0.25° \times 0.25°$ grid cells were counted to determine the sub-pixel heterogeneity. If the land cover type is identical for the corresponding model grid cell, the land cover type of this model grid cell is considered to be homogeneous. Model grid cells with multiple land cover types and over 60% grassland were defined as grassland-dominated mixed vegetation. Similarly, model grid cells
with mostly forest were classified as forest-dominated pixels. Grid cells with multiple different land covers were classified as mixed land cover. The forest cover of each $0.25° \times 0.25°$ grid cell was calculated with the percentage of forest (including evergreen, deciduous and mixed forest) pixels to investigate the impact of woody vegetation on soil moisture estimation.

## 2.3 Satellite observed water content

Satellite-observed near-surface soil moisture from SMOS and total water storage from GRACE were used in this study. GRACE
tracked the water movement from space by measuring the changes in the distance between the twin satellites caused by surface mass variations (Tapley et al., 2004). The JPL RL05M mass concentration (mascon) GRACE solutions (Watkins et al., 2015) were used to constrain model-simulated total water storage (i.e. the integration of surface water, soil water at three layers and groundwater stores). The GRACE data were represented on a $0.25°$ grid but they represent native resolution of $3° \times 3°$ equal-area caps. In contrast with sensing the integrated water content, SMOS characterizes global temporal change of
near-surface (0 - 5 cm) soil moisture from the microwave brightness temperature observations every three days (Kerr et al., 2010). The $0.25°$ Level-3 global daily soil moisture retrievals from CADTS (Centre Aval de Traitement des Données SMOS, https://www.catds.fr) (Jacquette et al., 2010; Kerr et al., 2013) for ascending and descending orbits were averaged over the overlapping area. The temporally and spatially varying uncertainties of GRACE and SMOS retrievals were provided as part of their respective products, and were used to investigate observation error variance-covariance matrices, $R$ (see Eq. 2 below),
in the assimilation method. The relative error was calculated as the ratio of the uncertainty over the absolute value for both GRACE and SMOS retrievals for each grid cell at each time step (Fig. 1).



## 2.4 International Soil Moisture Network

In-situ soil moisture observations at different depths available from the International Soil Moisture Network (ISMN) (Dorigo et al., 2011) were used to evaluate the performance of model-simulated soil moisture for the uppermost soil layer and root-zone. An additional level of quality control was imposed here on the ISMN data to eliminate those sites with less than 2 years data record, having persistently low or high values, or possessing inexplicable spikes or breaks in the time series. In total 385 stations from 19 measurement networks provided near-surface (0 – 5 cm) soil moisture observation globally, while 401 station from 15 networks provided root-zone soil moisture at 0 – 1 m (Fig. 2). Hourly observations were averaged over a 24-hour period to give daily moisture measurements. Stations with multiple measurements for soil moisture within 1 m depth were aggregated to soil moisture at 0–1 m. The 98th and 2nd percentiles of the data records for each site were assumed to represent the field capacity and wilting point required for the calculation of relative wetness.

## 2.5 Satellite observed greenness

The MODIS $0.05°$ monthly normalized difference vegetation index (NDVI) product (MOD13C2) (Didan, 2015) derived from atmospherically-corrected reflectance in red and near-infrared wavelengths were used as a simple, robust and well-known indicator for vegetation greenness. The MOD13C2 NDVI data were aggregated to $0.25°$ to be comparable with model simulations from January 2010 to December 2015. Areas of the Earth's surface that never exceeded a maximum NDVI value of 0.2 over this period were masked as barren land.

## 3 Method

### 3.1 Data assimilation

Satellite-derived total water storage and near-surface soil moisture were jointly assimilated into the global W3 model from 2010 to 2015. Systematic differences between model and observations need to be removed to ensure optimal performance of the assimilation method (Evensen, 1994; Dee, 2005; Renzullo et al., 2014). For the SMOS retrievals, our approach to eliminate systematic difference between model simulation and satellite observations of near-surface soil moisture was to convert both soil moisture values ($\theta_t$) and uncertainties into relative wetness ($w$) units using the field capacity ($\theta_{fc}$) and wilting point ($\theta_{wt}$) at each grid according to,

$$w_t = \frac{\theta_t - \theta_{wt}}{\theta_{fc} - \theta_{wt}} \tag{1}$$



and adjusting temporal mean and variance to that of W3 simulations. For total water storage, it was a simple matter of adding the W3 model simulated total water storage averaged over 2004 – 2009 to the GRACE observed water storage anomaly for absolute total water storage values.

Due to the disparity in temporal and spatial resolution and measurement depths between SMOS and GRACE, these contrasting
satellite water content observations were assimilated using an ensemble-based Kalman smoother approach with a one-month window, following the approach of Tian et al. (2017). Total water storage together with soil moisture data were used to constrain model-simulated water storage components formed as the state vector $x$, including surface water (rivers, lakes), vegetation water, soil water (top, shallow, and deep layer) and unconfined groundwater for each hydrological response unit. The observation vector $y$ consisted of the available daily SMOS surface soil moisture and the GRACE total water storage in a month at each
grid. Model error variance $P^f$ was derived from 100 ensemble members of the state variable, generated through the perturbation of precipitation, radiation and air temperature data. The analysis states $x^a$ were updated with the forecast states $x^f$ and the weighted difference between the observations and forecasts at the end of every month (Eq. 2), i.e.,

$$x_i^a = x_i^f + P^f H^T (H P^f H^T + R)^{-1} \left[ y - H(x_i^f) + \epsilon_i \right], \quad i = 1, \ldots, 100 \tag{2}$$

The matrix $P^f H^T (H P^f H^T + R)^{-1}$ above, known as the Kalman gain, determines the degree of influence that observation $y$
has on changing the model forecast state, $x^f$.

Spatially and temporally varying uncertainties from GRACE and SMOS products, characterised by $R$, were used in the assimilation to represent the observation error covariance matrix. Tian et al. (2017) applied an artificial weighting factor to the uncertainties of GRACE and SMOS data to compensate the over-adjustment from SMOS due to the inconsistency in units between SMOS and GRACE data. In this study, the first part of the observation operator $H$ converts SMOS soil moisture
retrievals in to available water content (in mm) for the upper most soil layer. The field capacity and wilting point from model simulations for the top 5 cm were applied to both soil wetness and uncertainties. No further weighting factor was required between GRACE and SMOS data after converting SMOS data to equivalent water height. Both ascending and descending SMOS soil moisture retrievals were used to improve the spatial coverage. The second part of the observation operator computes the monthly mean from the sum of daily water storage components in the state vector. The state variables for the next time step of
the model forward run were initialized with the analysed states.

The open-loop run (without assimilation of any observation), the assimilation of soil moisture alone and the assimilation of total water storage alone were also evaluated to examine the different impact on soil moisture profile adjustments. The same ensemble Kalman smoother was applied to the assimilation of SMOS alone (SMOS-only) and the assimilation of GRACE alone (GRACE-only) to compare with the joint assimilation. Since the uncertainty in SMOS data varies considerably between
land cover types, another joint assimilation experiment (Joint-landcover) was conducted where SMOS uncertainties were increased by 50% of the reported value over dense forest area (tree cover > 0.7) was implemented to identify any possible under-estimation of SMOS errors.



## 3.2 Evaluation of soil moisture estimates

Estimates of soil water content in the uppermost soil layer (0–5 cm) and shallow root-zone (0–1 m) after the joint assimilation were evaluated against in-situ soil moisture observations from ISMN. The in-situ stations within the corresponding model grid cell were aggregated to represent the soil moisture at $0.25°$ scale. The in-situ soil moisture monitoring sites were grouped based
on land cover type of the corresponding model grid cell. Both model-simulated and observed soil moisture were transformed to relative wetness to resolve differences in units and depths between model simulations and in-situ observations (Eq. 1). The performance of soil moisture estimation was statistically evaluated with Pearson correlation ($r$) and root-mean-square error (RMSE) for the open-loop and different assimilation experiments.

## 3.3 Analysis of vegetation response to root-zone soil moisture

Satellite-observed vegetation greenness can be used as an independent evaluation of root-zone soil moisture estimates, particularly in water-limited regions. The correlation of monthly NDVI and soil moisture estimates integrated over different depths after joint assimilation were calculated and compared with the open-loop results. The soil water storage estimates were integrated at four depths, namely, near-surface (0–5cm), shallow-root zone (0–1m), deep-root zone (0–10m) and total water column. The improvement in correlation was used as an indicator for enhanced performance. The correlation between monthly NDVI and
satellite-observed and model-simulated water content were also calculated for comparisons. The water content observations and estimates used in the comparisons included SMOS soil moisture, GRACE total water storage, model simulated root-zone soil moisture via the joint assimilation, and the precipitation-based soil moisture estimates from the antecedent precipitation index (API) with a constant decay coefficient of 0.9 (Hooke, 1979). API was used as it better represents the cumulative effects of precipitation on vegetation response than individual rainfall events.

Integrated soil water storage having maximum correlation with NDVI over time best reflects the surface greenness and vegetation condition. The plant-available soil water storage were simply determined globally with the correlation between integrated soil water storage and NDVI. Annual linear trends in this variable were calculated to determine the area under soil water stress and with potential for vegetation decline. Linear trend analysis was also applied to the annual average NDVI to investigate the consistency between vegetation greenness and soil water storage. The trends in soil water storage derived from the open-loop
and joint assimilation were compared with the trends in NDVI to investigate the impacts of joint assimilation on annual trends.



## 4  Results

### 4.1  Near-surface and root-zone soil moisture estimation

SMOS soil wetness and W3 top-layer soil wetness from open-loop and data assimilation were compared with the in-situ near-surface soil wetness observations from ISMN (Fig. 3). Satellite observations of soil moisture (SMOS) were generally better

correlated with in-situ soil moisture observations over non-forest areas than open-loop simulations (Fig. 3a). However, as the fraction of tree cover increases, the relative performance changes and model simulations tend to be better correlated with in-situ measurements than SMOS. The joint assimilation of both SMOS and GRACE observations (Fig. 3b) shows improved correlation with in-situ measurements compared with the model open-loop over the majority of the non-forest sites (i.e. more points above the 1-to-1 line). This overall improvement is due to data assimilation bringing the model and SMOS soil moisture

into better agreement for these sites, as illustrated in Fig. 3c for a grassland site. The improved agreement with SMOS retrievals for forest sites corresponds to a decrease in correlation with in-situ measurements (Fig. 3d).

The impact of data assimilation on W3 model performance is further illustrated in Fig. 4. For near surface soil moisture (Fig. 4a), data assimilation improved correlation and reduced RMSE against ISMN measurements. Joint assimilation of SMOS and GRACE yielded better results than the assimilation of SMOS or GRACE alone, and joint assimilation results were, on average,

best and marginally better than SMOS-only results. The assimilation of GRACE data alone had little impact on surface soil moisture estimation. Further improvement to the joint assimilation was observed when the influences of the SMOS observations were down-weighted for ISMN sites in forest (high tree cover) areas (plot labeled 'Joint-landcover').

Data assimilation resulted in significant improvements in W3 root-zone soil moisture estimation over the majority of sites (Fig. 4b). In contrast to surface soil wetness, SMOS and GRACE observation both impacted deeper soil wetness estimation

considerably. Simulation of soil wetness over the root-zone (0–1m) was less affected by forest cover compared to the near-surface soil wetness, as evident from the high degree of similarity between the 'Joint' and 'Joint-landcover' plots (Fig. 4b). This suggests no significant difference in performance as a result of down-weighting SMOS influence over forest regions.

Table 1 summarises W3 model soil moisture estimation performance for both near-surface and the root-zone for different land cover types. Overall model uncertainty (i.e. ensemble spread) decreased significantly after the assimilation for all land cover

types. On average, the correlation with in-situ observations increased for both surface and root-zone soil moisture estimates. The improvements in surface soil moisture estimates were mainly over grassland, cropland and grassland dominated areas, with changes in correlation, $r^a - r^o$, as high as 0.36 for grassland. Correlation in model surface soil moisture estimates over savannas and forest areas decreased relative to open-loop simulations. Data assimilation improved root-zone soil moisture estimates for most land cover types. The exceptions were savannas, evergreen and deciduous forests, with up to 0.75 increase

over mix-types areas and an average change in correlation of $\sim 0.1$.



## 4.2 Relation between vegetation greenness and soil water availability

Monthly water storage integrated to different depths, from the uppermost soil layer to total soil column, were compared with satellite observed greenness. The response of vegetation greenness to water storage at different depths is illustrated for selected sites over six land cover types in Fig. 5. Significant differences in soil water variability were found between the joint assimilation and open-loop estimates in all sites (Fig.5), in particular deep-root zone and total water column. The temporal pattern in greenness and water storage time series was characterised for open-loop and joint assimilation estimates by correlation, $r^o$ and $r^a$ respectively. As an example, the grassland site in northern China responded more strongly to the availability of near-surface soil water (higher correlation) than deep soil water and total water storage (Fig. 5a). This suggests a shorter time lag between surface soil water availability and surface greenness. Stronger correlations between NDVI and near surface soil water storage were found increased by 0.15 after assimilation (i.e. $r^a - r^o = 0.15$). Greenness of the savannas site in eastern Brazil and the cropland site in southern Australia showed similar seasonal pattern (correlation) to water storage over all depths (Fig. 5b and 5c). The largest change in correlation as a result of joint assimilation was observed for the Brazil savannas site (Fig. 5b) ($r^a - r^o > 0.4$) for shallow- and deep- water storage. NDVI in shrublands and forest sites with deeper roots showed higher correlation with deep soil water and total water storage availability, such as the evergreen broadleaf forest in Nigeria, shrubland in northern Mexico and deciduous broadleaf forest in southern Bolivia (Fig. 5d to 5f).

Significant increases in correlation between W3 water storage and vegetation greenness resulted from the joint assimilation of SMOS and GRACE. Changes in correlation, compared to open-loop simulations, between monthly storages estimates at different depths and satellite-derived greenness are given in Table 2 for different land cover types. Correlation between grassland greenness and soil water availability in the near-surface through to the total water column increased by 0.1 to 0.47 on average. Cropland NDVI was better correlated with soil water availability near-surface layer from joint assimilation than the W3 open-loop by 0.1. Greater increases in correlation were found for the deeper water storage compared to shallow soil water as a result of data assimilation, particularly for open shrubland, savannas, wetland and forest dominated regions. Marginal changes were found over the closed shrubland, evergreen broad leaf forest and snow-dominated regions.

Fig. 6 shows the maximum change in correlation as a result of joint assimilation. Correlations were computed between NDVI and all possible W3 storage depths. Increases in correlations between soil water availability and NDVI with the joint assimilation were observed globally, most notably in the high latitudes of the northern hemisphere, where increases in correlation as high as 0.5 were widespread. This is due to the joint assimilation bringing the seasonality of soil water availability into better agreement with greenness.

Having established that joint assimilation improved soil water estimation and the correlation with vegetation response globally, we explored the sources of the improvements. The correlation between NDVI and soil water content estimates from API, SMOS, GRACE and W3 were computed globally (Fig. 7). API showed a correlation with NDVI of $\sim 0.5 - 0.6$ over major dry lands and high latitude regions, except for western and southern Australia and Europe (Fig. 7a). Near-surface soil moisture



estimates from SMOS showed strong positive correlation with vegetation conditions over tropical grassland and savannas regions, but strong negative correlation over eastern America and Europe (Fig. 7b). Vegetation growth over tropical regions showed clear wet and dry seasons patterns closely related to the variability of total water storage from GRACE (Fig. 7c). The map based on the correlation of soil water storage (from joint assimilation) for the integration depth with the highest correlation with NDVI (Fig. 7d) shows the strong correlation between available root-zone soil water and vegetation greenness in the semi-arid and arid regions compared to other water content estimates.

## 4.3 Trends in soil water availability and vegetation response

The change of plant-available soil water storage estimated from the W3 open-loop and joint assimilation from January 2010 to December 2015 was compared with the change of global vegetation greenness over the same period and clear differences in the magnitude of soil water storage change and high spatial variability were observed globally (Fig. 8a and 8b). For example, a decrease of 10 mm/yr in soil water storage was simulated in open-loop simulations over northern Mexico, central Africa and northeastern China. However joint assimilation results showed an increase in soil water storage for these same regions. Differences in water storage change between open-loop and joint assimilation (Fig. 8c) could be as large as 20 mm/yr, and were most noticeable over southeast Asia and the majority of South America.

Clear decreasing trends in NDVI (more than 0.025 units per year) were observed over central and eastern Australia (Fig. 8d), while decreasing trends in soil water availability of over 15 mm/yr were found in both model open-loop and joint assimilation estimates. A greater decrease of soil water storage was inferred in central and eastern Australia through joint assimilation than from the open-loop. Similarly, the deficit in plant-available soil water storage estimated through joint assimilation aligned well with the dramatic decrease in vegetation greenness in eastern Brazil, southern India and southern Africa. The joint assimilation resulted in estimated increases in soil water storage that were globally much more consistent with increased greenness than the open-loop.

## 5 Discussion

We found that global modelling of root-zone soil moisture can be improved substantially through the joint assimilation of GRACE total water storage and SMOS soil moisture retrievals. This is consistent with previous findings for Australia (Tian et al., 2017). Corrections to near-surface soil moisture estimates resulted mainly from the assimilation of SMOS soil moisture. Uncertainties in SMOS soil moisture retrievals, e.g. related to errors in surface roughness and vegetation cover characterisation, influenced the accuracy of the estimation through the weights of the Kalman gain (Eq. 2). Therefore, in locations where satellite soil moisture estimates were observed to be more accurate than the W3 model open-loop simulations (e.g. grassland and cropland areas), the assimilation of these data improved the agreement between model estimates of near-surface soil moisture





and in-situ observations. However, assimilation of SMOS degraded model estimation for grid cells dominated by forest or mixed land cover types, most likely due to the underestimated SMOS errors. The SMOS relative errors for each cover type ranged from 7–15% at median values, but was as high as 50% for full forest coverage regions (Fig. 1a). The average relative errors for SMOS over 50% of the in-situ sites with mixed land cover types were only 9% (Table 1), indicating the under-

estimation of SMOS error in those grid cells. By increasing the error of SMOS observations for forest areas, thus reducing their influence on assimilation, the number of sites with degraded model estimation was reduced (Fig. 4a). This suggests that reported errors associated with the SMOS product are likely underestimated for densely vegetated areas.

The suitability of ISMN in-situ soil moisture measurements for evaluation needs to be considered in interpretation. The quality of the data records, the sparseness of network coverage, uneven distribution globally (e.g. heavily skewed to North America), as

well as the representativeness of a single site, or very small number of sites within a model or satellite pixel ($\sim 0.25° \times 0.25°$) are important contributing factors to the evaluation statistics. By careful inspection of the in-situ data and removal of any sites with insufficient data and time series with unrealistic behaviour, model evaluation could be conducted with a subset of 77% of the full complement of ISMN data that was believed to be of better quality. Given the paramount importance of these data in evaluation of model and satellite products in general, it is critical that the ISMN and similar in-situ measurement networks are

maintained, but rigorous quality control is equally important.

The assimilation of GRACE data had marginal impact on W3 near-surface soil moisture simulation (Fig. 4). In contrast to the SMOS product, uncertainties in GRACE data were less variable in terms of relative error across land cover type, with error between 10–15% on average (Fig. 1b). A majority of the modelling grid cells showed improved correlation and reduced RMSE of root-zone soil moisture as a results of joint assimilation of GRACE and SMOS, not only in the grassland- dominated sites

but also for mixed land cover types. The improved root-zone soil water estimation in the joint assimilation could be linked to the Kalman smoother, which used the SMOS (daily) data to temporally disaggregate the GRACE observed (monthly) total water storage. Therefore, not only does the joint assimilation of SMOS and GRACE observations vertically redistribute the water storage change into different W3 soil layers, it also redistributes the change temporally based on different dynamics of the soil moisture signal at the different depths.

Root-zone soil moisture varies considerably in space, as do plant rooting depth and soil physical properties. This makes it a challenge to compare model estimates over a cell and in-situ measurements at point scale. Remotely sensed vegetation greenness can serve as a surrogate for water availability in water-limited regions of the world. MODIS NDVI was used as an independent dataset to evaluate root-zone soil moisture simulations. Significant increases in correlation were found globally after joint assimilation (Fig. 6). The increased correlation between water availability and vegetation condition in semi-arid to

arid regions is encouraging as it may result in improved capability for forecasting drought and vegetation productivity. The improvements over temperate and polar zones are more likely due to better consistency with NDVI seasonality that is unrelated to water-limitation. The correlation between deep-root zone soil water storage and NDVI were found to have the greatest improvement with an increase of over 0.47 on average for grasslands and 0.35 for shrublands (Table 2). This is likely due to



the time lag between vegetation response to precipitation and surface soil moisture (Chen et al., 2014). Deeper soil moisture constrained by GRACE observations has a longer memory and incorporates climate conditions of the preceding months. The greater improvements of cropland NDVI correlation with near-surface soil moisture indicates the benefits of integrating SMOS data for better characterisation of soil moisture dynamics.

The response of vegetation to water availability at different depths varies according to vegetation type and climate. For example, grasslands over the western U.S. and northeastern China showed strong correlation with SMOS near-surface soil moisture retrievals and modeled surface soil moisture, but week correlation with GRACE observed total water storage (Fig. 5a and 7). On the other hand, grassland in Sahel showed the same relative response to water availability at different depths but higher correlation with deep soil availability (Fig. 7). This appears to be due to the relatively deep root zone and lesser water hold-

ing capacity (Leenaars et al., 2018). Identifying the soil layers that contribute most to the temporal behaviour of vegetation greenness is critical for understanding the impacts of water stress on the terrestrial ecosystem. Fig. 7d shows the correlations between the vegetation conditions and the soil water storage in the corresponding integrated soil water layer. Soil water availability strongly reflected vegetation conditions over most of the globe except for the Amazon, north America and Europe (Fig. 7d). Low correlation over the Amazon rain forest was expected, since it is primarily energy-limited (Nemani et al., 2003). The

SMOS and GRACE observations both showed negative correlation with the surface greenness over Europe and eastern North America, where better correlations were found with precipitation based index (API, in Fig. 7a). This indicates that vegetation over these regions quickly responds to precipitation. The time lag in vegetation response to change in soil water availability over these regions, as well as the limitations of radiation and temperature may also explain the negative correlation (Wu et al., 2015). Overall, the soil water storage derived from the joint assimilation embodied the best knowledge of available water con-

tent not only from meteorological forcing data, but also from the SMOS near-surface soil moisture and GRACE total water storage. Given accurate information of soil water availability, vegetation vigor and productivity can potentially be predicted (Tian et al., submitted).

A number of severe droughts have occurred during the last decade, including the droughts in Sahel, East Africa, California, China and northeastern Australia. The annual trends in NDVI and soil water storage from January 2010 to December 2015

showed consistency in the spatial patterns between surface greenness and soil water availability during and after the drought conditions. After a sharp recovery from the Millennium drought during an extremely wet period from 2010 to 2011 (Leblanc et al., 2009; van Dijk et al., 2013a), drought returned to eastern Australia with a decrease in soil water over 15 mm/yr estimated from both model open-loop and joint assimilation (Fig. 8a and 8b). A decline in NDVI of more than 0.025 units per year was observed for the majority of middle and eastern Australia due to the developing soil water deficit (Fig. 8d). Increases in soil

water deficit were enhanced as a result of assimilating GRACE and SMOS over eastern Brazil, California and south Africa and this was consistent with a decrease in vegetation greenness in these areas. The stronger signal of water storage deficiency compared to the open-loop is mainly attributed to GRACE-observed decreasing total water storage in agreement with Rodell et al. (2018). The severity of groundwater depletion for irrigation in northern and southern India, as observed by GRACE (Rodell et al., 2009), was also better captured by through the assimilation of GRACE (Fig. 8a). Joint assimilation of GRACE



and SMOS sometimes reversed the direction of change in soil water storage, compared to the open-loop, resulting in better agreement with trends of temporal pattern in NDVI, particularly in central Africa, Mexico, and eastern China.

## 6 Conclusions

This work has demonstrated that the joint assimilation of GRACE and SMOS data into an ecohydrological model resulted in a
spatial and temporal redistribution of water storage that significantly improved root-zone soil moisture estimation over different land cover types globally. In particular, improvements were found for estimation of shallow and deep soil water availability. The joint assimilation optimally integrated the water dynamics information from SMOS and GRACE and mitigated the deficiencies of the individual sources of observation.

Vegetation response to soil water availability at different depths was found to vary according to ecosystem and climate. The
close relationship between vegetation growth and soil water availability was quantified firstly with the root-zone soil water estimates through the assimilation of satellite soil moisture and total water storage retrievals simultaneously. The improved agreement between vegetation vigor and soil water availability indicates the potential for improving ecohydrological models and anticipating changes in vegetation condition. Accurate characterization of vegetation response to soil water availability also provides new insights to help improve monitoring and forecasting drought impacts on ecosystems.

*Data availability.* The ecohydrological model W3 is available online at http://wald.anu.science. The MOD13C2 data was retrieved from online Data Pool, courtesy of the NASA EOSDIS Land Processes Distributed Active Archive Center (LP DAAC), USGS/Earth Resources Observation and Science (EROS) Center, Sioux Falls, South Dakota, https://lpdaac.usgs.gov. GRACE land mascon solutions are available at http://grace.jpl.nasa.gov, supported by the NASA MEaSUREs Program. The CATDS level-3 daily soil moisture retrievals is available at https://www.catds.fr/sipad/.

*Competing interests.* There are no competing interests.

*Acknowledgements.* This research was supported through ARC Discovery grant DP140103679. This research was undertaken with the assistance of resources and services from the National Computational Infrastructure (NCI), which is supported by the Australian Government.




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





**Figure 1.** Averaged relative error of satellite observed water content in different land cover types for: (a) SMOS-derived soil moisture; (b) GRACE-derived total water storage.





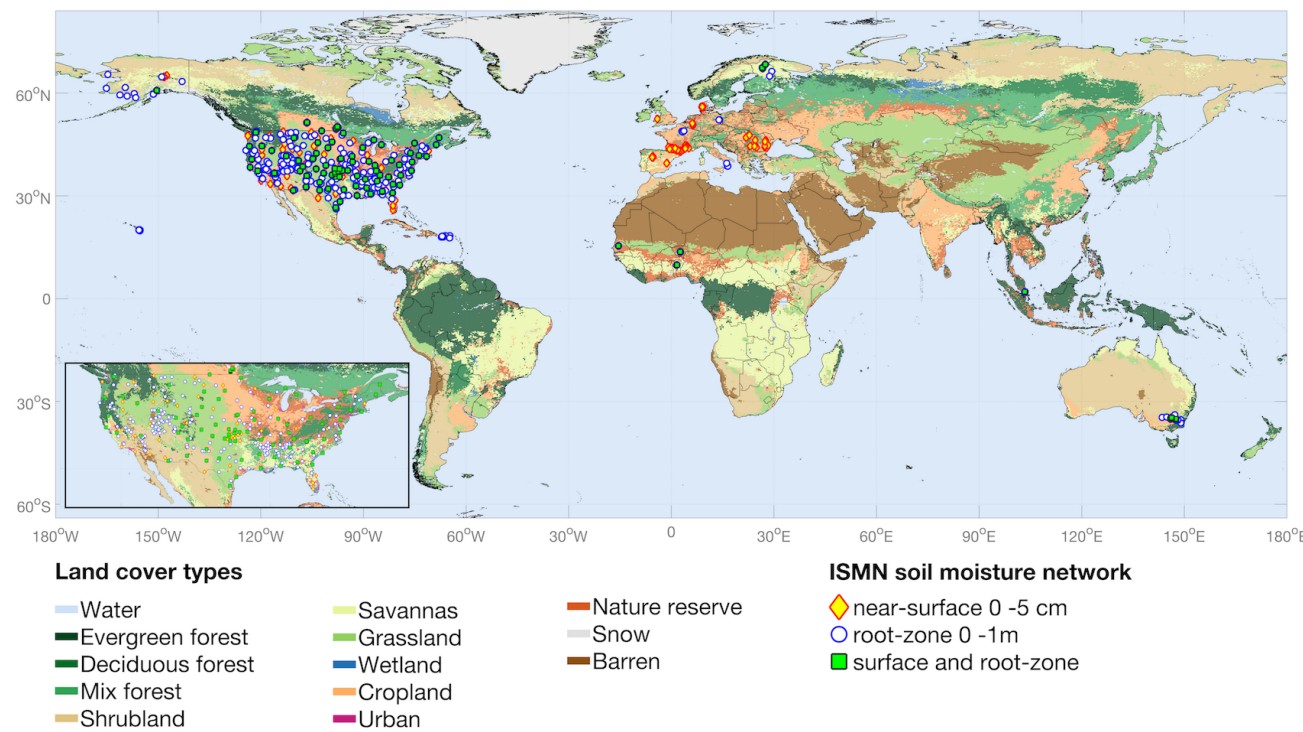

**Figure 2.** Distribution of in-situ near-surface and root-zone soil moisture sites from the International Soil Moisture Network (ISMN) overlaid on the background of MODIS IGBP (International Geosphere–Biosphere Programme) land cover classifications (MCD12Q1).





**Figure 3.** Assessment of near-surface soil moisture estimation with ISMN in-situ measurements over the six-year period from 2010 to 2015: (a) correlations of SMOS soil moisture retrievals with in-situ measurements (y-axis) compared against open-loop (x-axis); (b) correlation of near soil moisture estimates after the joint assimilation with in-situ measurements (y-axis) compared against model open-loop (x-axis) ; Each ISMN site is characterised by the fraction of tree cover within the corresponding $0.25°$ cell. (c) and (d) time series of simulated surface soil moisture before and after the joint assimilation over grassland and forest dominated region.





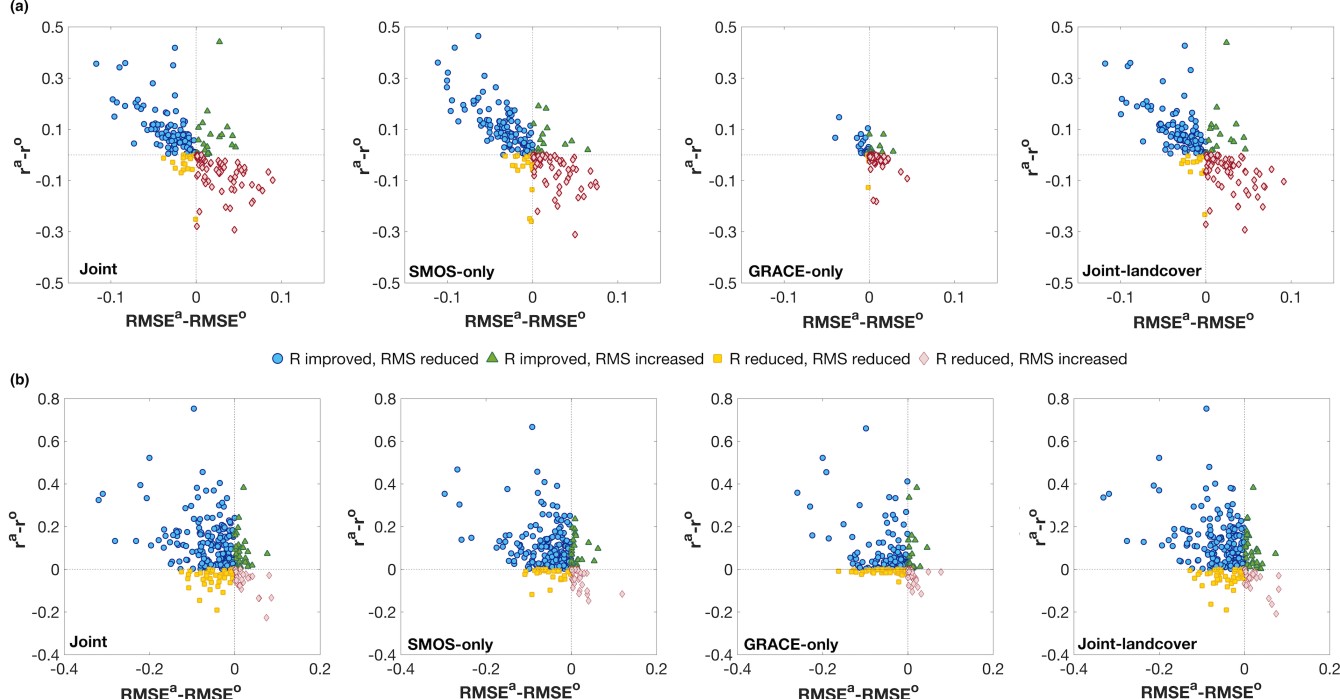

**Figure 4.** Performance of surface and root-zone soil moisture estimates from four data assimilation scenarios against open-loop: correlation and root mean-squared error (RMSE) change ($r^a - r^o$, RMSE$^a$-RMSE$^o$) after the assimilation in (a) surface soil moisture estimation; (b) root-zone soil moisture estimation. The four scenarios include: Joint as joint assimilation of SMOS and GRACE, SMOS-only as the assimilation of SMOS data alone, GRACE-only as the assimilation of GRACE data only, Joint-landcover as increasing SMOS uncertainty in forest regions in the joint assimilation. The points in the scatter plots are colour coded such that: blue indicates ISMN sites where improvement was observed in both correlation and RMSE; green indicates sites where there was improvement in correlation, but not in RMSE; yellow indicates those sites where there was improved RMSE, but not correlation; and red indicating sites where assimilation resulted in degradation in both correlation and RMSE.





**Figure 5.** Time series of vegetation responses (NDVI) to soil water storage over different integrated depths across land vegetation types. Also displayed are the correlation between soil water storage and NDVI (top plots) for open loop ($r^o$, red curves) and joint assimilation ($r^a$, blue curves).



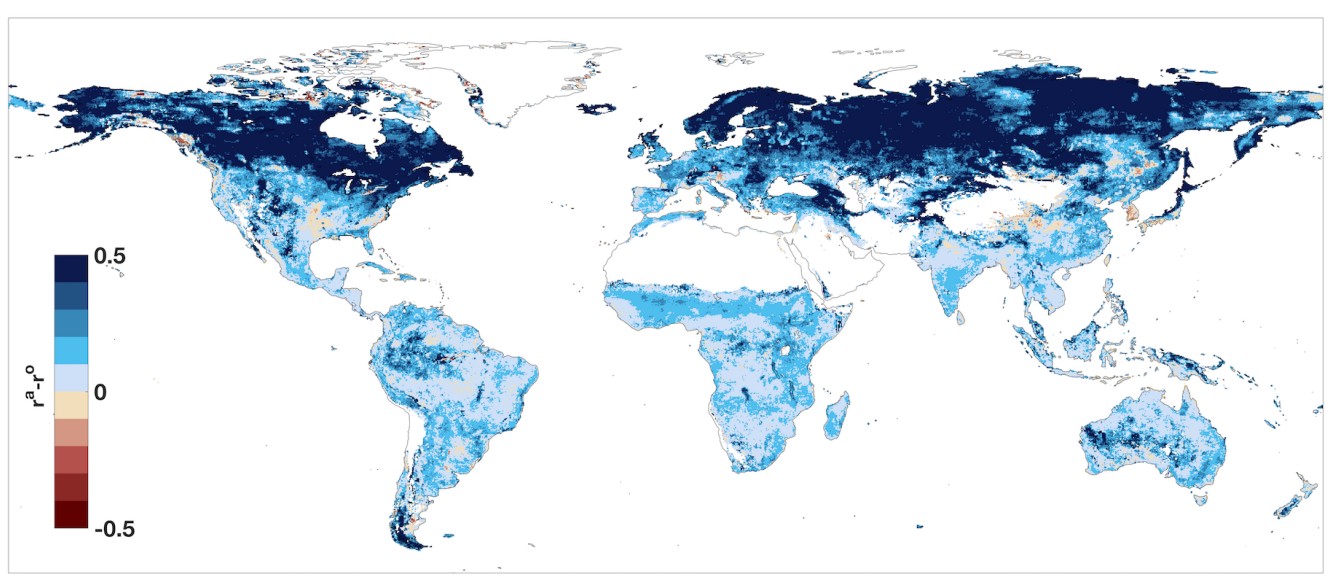

**Figure 6.** Maximum change in correlation $(r^a - r^o)$ of satellite observed monthly vegetation greenness and soil water storage over different integrated depths estimated through the joint assimilation $(r^a)$ and model open-loop $(r^o)$.



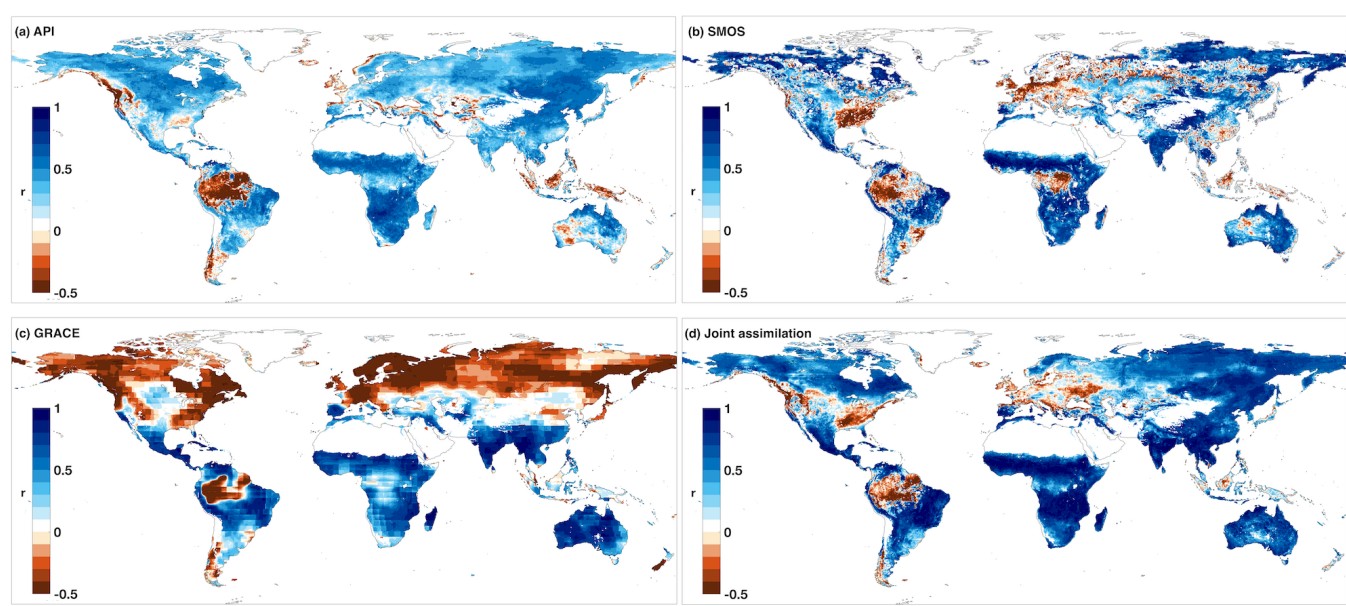

**Figure 7.** Vegetation response to different sources of soil water availability as indicated by: the correlation between monthly NDVI and (a) antecedent precipitation index (API), (b) SMOS surface soil moisture retrievals, (c) GRACE total water storage change retrievals, and (d) plant available soil water storage derived through the joint assimilation.





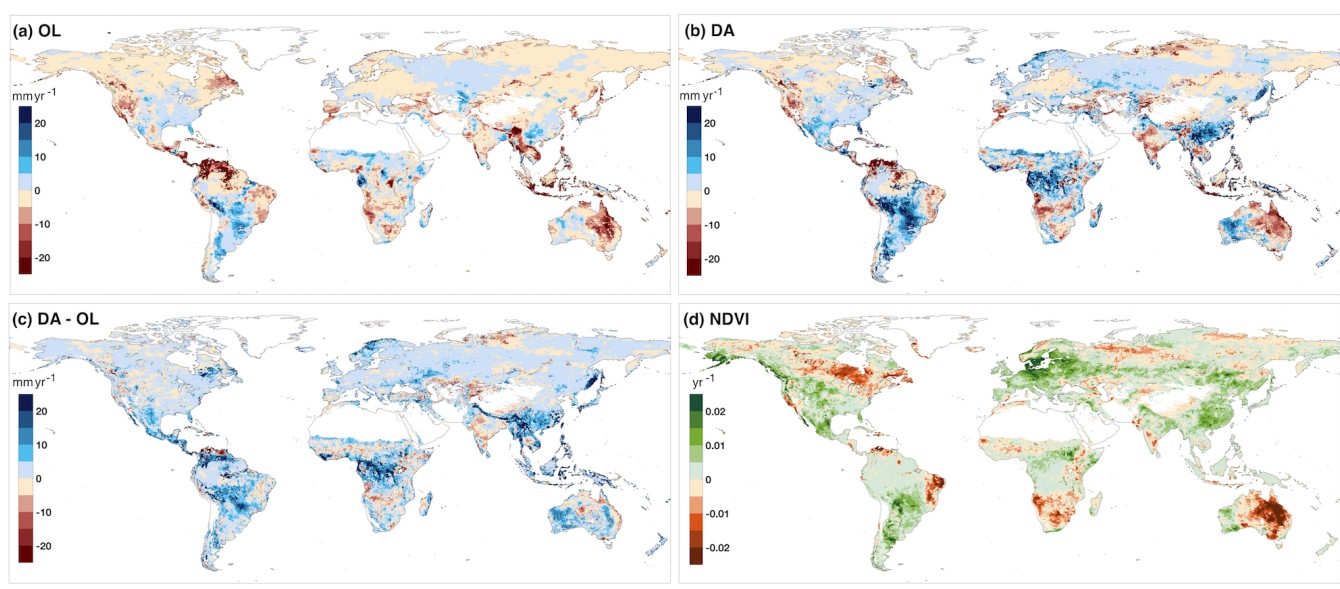

**Figure 8.** Change of plant available soil water storage and its impacts on vegetation greenness from 2010 to 2015: linear trends of available soil water storage estimated from (a) model open-loop and (b) joint assimilation; (c) difference in trends between joint assimilation and model open-loop; (d) linear trends of NDVI.





**Table 1.** Evaluation of near-surface and root-zone soil moisture estimation with ISMN in-situ soil moisture observation across land cover types

### near-surface soil moisture 0-5cm

|  | GS | OS | WS | SS | CP | CV | EN | DB | MF | GD | FD | ML |
|---|---|---|---|---|---|---|---|---|---|---|---|---|
| Number of grid cells | 35 | 6 | 7 | 2 | 35 | 4 | 3 | 1 | 6 | 28 | 8 | 68 |
| Average SMOS uncertainty | 6% | 8% | 12% | 7% | 6% | 11% | 11% | 7% | 17% | 7% | 24% | 9% |
| Average GRACE uncertainty | 9% | 12% | 12% | 6% | 9% | 9% | 5% | 10% | 9% | 9% | 7% | 10% |
| Average Model OL uncertainty | 11% | 10% | 9% | 9% | 11% | 9% | 7% | 7% | 8% | 10% | 8% | 9% |
| Average Joint DA uncertainty | 4% | 3% | 5% | 3% | 3% | 4% | 4% | 5% | 4% | 3% | 5% | 4% |
| Average open-loop correlation $r^o$ | 0.58 | 0.65 | 0.58 | 0.85 | 0.51 | 0.56 | 0.57 | 0.36 | 0.58 | 0.60 | 0.62 | 0.60 |
| Max correlation change $r^a - r^o$ | 0.36 | 0.03 | 0.11 | -0.12 | 0.44 | 0.12 | 0.03 | -0.08 | 0.12 | 0.22 | 0.06 | 0.28 |
| Average correlation change $r^a - r^o$ | 0.07 | -0.04 | -0.02 | -0.15 | 0.08 | 0.04 | 0 | -0.08 | 0.01 | 0.06 | -0.01 | -0.01 |

### root-zone soil moisture 0-1m

|  | GS | OS | WS | SS | CP | CV | EN | DB | MF | GD | FD | ML |
|---|---|---|---|---|---|---|---|---|---|---|---|---|
| Number of grid cells | 51 | 11 | 6 | 2 | 30 | 9 | 4 | 2 | 4 | 42 | 19 | 106 |
| Average SMOS uncertainty | 6% | 8% | 12% | 7% | 5% | 10% | 11% | 8% | 13% | 7% | 18% | 9% |
| Average GRACE uncertainty | 9% | 9% | 9% | 6% | 7% | 8% | 6% | 8% | 9% | 8% | 8% | 10% |
| Average Model OL uncertainty | 11% | 9% | 9% | 9% | 10% | 8% | 6% | 8% | 8% | 10% | 8% | 9% |
| Average Joint DA uncertainty | 4% | 3% | 5% | 3% | 4% | 5% | 2% | 5% | 4% | 4% | 5% | 5% |
| Average open-loop correlation $r^o$ | 0.36 | 0.34 | 0.39 | 0.89 | 0.47 | 0.64 | 0.76 | 0.65 | 0.25 | 0.48 | 0.51 | 0.45 |
| Max correlation change $r^a - r^o$ | 0.46 | 0.37 | 0.18 | 0 | 0.31 | 0.16 | 0.1 | -0.04 | 0.17 | 0.39 | 0.17 | 0.75 |
| Average correlation change $r^a - r^o$ | 0.10 | 0.04 | 0.09 | -0.07 | 0.12 | 0.03 | 0.01 | -0.05 | 0.08 | 0.07 | 0.04 | 0.08 |

Land cover types:
GS: grassland OS: open shrubland WS: woody savannas SS: savannas CP: cropland CV: cropland/natural vegetation EN: evergreen needle leaf forest
DB: deciduous broad leaf forest MF: mix forest GD: grassland-dominated mix types FD: forest dominated mix types ML: mixed land covers





**Table 2.** Average changes in correlation ($r^a - r^o$) of water storage estimates integrated at different depths against satellite observed vegetation greenness for different land cover types

| | GS | CS | OS | WS | SS | CP | CV | EN | EB | DN | DB | MF | BN | UN | SN | WT | GD | FD |
|---|---|---|---|---|---|---|---|---|---|---|---|---|---|---|---|---|---|---|
| Near-surface | 0.10 | 0.01 | 0.04 | 0.14 | 0.10 | 0.09 | 0.03 | 0.05 | 0.05 | 0.04 | 0.09 | 0.11 | -0.03 | 0.12 | -0.05 | 0.10 | 0.07 | 0.07 |
| Shallow-root zone | 0.22 | 0.07 | 0.27 | 0.06 | 0.18 | -0.04 | 0.07 | 0.06 | 0.02 | 0.12 | 0.10 | 0.10 | -0.03 | 0.10 | -0.02 | 0.17 | 0.11 | 0.14 |
| Deep-root zone | 0.32 | 0.05 | 0.34 | 0.11 | 0.28 | 0.02 | 0.18 | 0.09 | -0.02 | 0.15 | 0.13 | 0.13 | -0.08 | 0.12 | 0.01 | 0.29 | 0.14 | 0.20 |
| Total water column | 0.47 | 0.05 | 0.35 | 0.14 | 0.35 | 0.02 | 0.20 | 0.13 | -0.03 | 0.14 | 0.11 | 0.12 | -0.08 | 0.13 | 0.01 | 0.46 | 0.13 | 0.25 |

Land cover types:

GS: grassland CS: closed shrubland OS: open shrubland WS: woody savannas SS: savannas CP: cropland CV: cropland/natural vegetation EN: evergreen needle leaf forest
EB: evergreen broad leaf DN: deciduous needle lead DB: deciduous broad leaf forest MF: mix forest BN: barren UN: urban SN: snow WT: wetland GD: grassland-dominated mix types FD: forest dominated mix types