# Peer review of "Global joint assimilation of GRACE and SMOS for improved estimation of root-zone soil moisture and vegetation response"

_Hydrology and Earth System Sciences, 2018_

## Referee Comment (RC1) · Anonymous Referee #1 · 24 Sep 2018

This manuscript presents the results of assimilating SMOS and GRACE data, jointly and independently, into an eco-hydrological model at the global scale. Model results were evaluated using in situ soil moisture and MODIS NDVI estimates. The assimilation method seems sound and the results are reasonable and should be of great interest to researchers in the community. My comments are mainly on how the assimilation impacts are assessed using NDVI. The authors employed MODIS NDVI to assess the impact of data assimilation on model estimates and suggested that increased correlation between soil moisture and NDVI is an indication of improved water storage estimates. This assumption may only be valid in arid and semi-arid regions; in other regions where ET is limited by available energy, changes in NDVI and water storage may

[Figure]

Creative Commons BY license logo

not be correlated at all. I think the correlation presented in the paper may have been affected by the seasonality in the data which is easy to see in Fig.5 where increase in correlation is associated with increased seasonal amplitudes in model estimates. If the intention is to report on increased seasonality, it may be better to compute the changes in seasonality; otherwise, I would suggest the authors compute the correlation based on anomalies relative to monthly mean to eliminate the influence of seasonal variation. In either case, physical-based evidence and reasoning needs to be provided for justifying the use of NDVI in this purpose. In fact, the authors may look into the correlation between in situ root zone soil moisture and MODIS NDVI to see if and when this assumption is correct. In addition, if NDVI is truly correlated with water storage, wouldn't comparing model estimated ET with NDVI make more sense? The manuscript can also be improved with more details on the model and the rationale behind pre-processing satellite data. For instance, the ecological aspect of the model is never described and no results were provided on how vegetation responded to data assimilation. And there is no discussion on why SMOS data needed to be scaled before assimilation while GRACE data were not. What are the adverse impacts of assimilating SMOS without scaling and how are they related to any model behaviors?

Additional comments: Page 1 Line 12: Increased correlation between vegetation greenness and soil water storage does not necessarily mean improvement on water storage estimates, unless you back this up with in situ observations. Line 21: root depths do not unilaterally determine whether plants can access deeper water stores. The capillary force can also lift water up from deeper water stores to near surface soils.

Page 2 Line 23-25: Vegetation water content only constitutes a very small part of GRACE derived TWS and thus the lag may not be related to vegetation greenness at all.

Page 3 Section 2.1. Does groundwater interact with soil moisture?

Page 4 Line 18: a 0.25 grid or 0.5? Line 25: Is absolute value the total error of

SMOS and GRACE? What is the purpose for calculating relative errors base on land cover types (i.e., Fig. 1)? Given the coarser scale of GRACE data and the fact that vegetation water content is only a small part of total water storage, I don't think GRACE errors are related to vegetation types.

Page 5 Line 9-10: Using the 2nd and 98th percentiles as wilting point and field capacity can be a problem if the sites are located in very drier or wet climate where soil moisture is often restricted to one side of its full range. Line 23: Were the field capacity and wilting point here derived from SMOS measurements or from the model? After the adjustment, would you convert relative wetness back to soil moisture content for assimilation?

Page 6 Line 1: Adjusting temporal variance of SMOS data is not bias correction. Why did you need to do that? And why didn't you adjust the temporal variance of GRACE data to match that of W3 estimates? Line 13: equation (2). How does this update work? SMOS and GRACE have different spatial and temporal resolutions? Line 20: why do you need field capacity and wilting point to convert soil moisture content to equivalent water heights?

Page 7 Line 14: Do the two "correlation" words mean the same thing? Line 17: The API needs to be introduced in the data section. Line 20: This sentence does not sound right. In fact, I find this whole paragraph difficult to understand.

Page 8 Line 4: It looks to me that Fig. 3(b) has more points below the 1-1 line. Can you provide average r for DA and open loop, respectively? Line 9-11: joint assimilation improves SMOS soil moisture retrievals? Line 13: Fig. 4. Why are there fewer data points for the GRACE-alone plot? Line 16: This further improvement is not obvious to me. You probably should provide averaged statistics in numbers to back this up. Line 20: "less affected" is not obvious to me. Line 23: Listing uncertainties of SMOS and GRACE in Table 1 can be mis-leading as they are not relative to the same set of in situ data and have nothing to do with improvements made by data assimilation. Besides,

they are not referred to throughout the paper.

Page 9 Section 4.2. I don't think increased correlation with NDVI can be counted as improvements. In many cases, it is due to increased seasonality such as in Savannas, eastern Brazil and hence increased correlation with NDVI which has strong seasonal changes. correlation in Figs. 6 and 7 should be calculated based on anomalies relative to monthly mean; otherwise, it reflects mostly the seasonality.

Page 10 Line 8: what is exactly plant-available soil moisture storage? Root zone soil moisture? Section 4.3. Trend should be calculated using anomalies relative to monthly mean. Line 4: Can you include the correlation between soil water storage from the open loop with NDVI in Fig. 7?

Page 11 Line 27. You stated earlier that "greenness can serve as a surrogate for water availability in water-limited regions". But NDVI was used to evaluate changes in water storage across the globe, regardless of climate conditions.

---

## Referee Comment (RC2) · Anonymous Referee #2 · 20 Oct 2018

The submitted manuscript by Siyuan Tian et al investigated the impacts of assimilating satellite water content retrievals on the estimation of surface and root-zone soil moisture over the globe and across different land cover types. The authors aimed at improving the accuracy of root-zone soil moisture prediction by jointly assimilating satellite-observed soil moisture from SMOS and total water storage changes from GRACE into a global ecohydrological model. They then evaluated the performance of the joint assimilation by comparing against the open-loop model and alternative assimilation methods with ground-based soil moisture measurements and vegetation index.

This paper is well written, properly structured and presented, with interesting results

being thoroughly interpreted by a good discussion. I believe this manuscript will be interesting to future HESS readers and contribute to the international literature. There are two major concerns that I would like the authors to address before the publication of this manuscript.

1. While GRACE-derived TWSA provides an integrated measurement of water storage changes above and underneath the earth surface, why would near-surface soil moisture derived from SMOS still be required? Don't SMOS and GRACE monitor overlap water content at near-surface? This has not been fully justified and explained in the Introduction or in the ecohydrological modelling method.

2. Following up Reviewer#1's major comment on assessing assimilated soil moisture using NDVI, I do agree Reviewer#1 that extra experiments of correlation analyses based on de-seasonalized times series of all data are required. Although I agree with the authors that the improvements of the modelled root-zone soil moisture over only ET limited regions are likely due to increased seasonality, authors may need to show how the methods proposed in this study could improve root-zone soil moisture in the long run without the effect of seasonality.

My specific comments are as follows:

1). Page1, Line 4:Do you have references to confirm this? Some people believe GRACE-derived TWSA is mainly dominated by soil moisture variation over many places.

2). Page3, Line 9-18: Introduction is well presented, however, this paragraph of objectives could be improved by clearly numbering each objective such as 1)…. 2)….3)…. This will make it easier for future readers to get straight to the points.

3). Page3, Line 27: includes —> including, and these .

4). Page3, Line 21-30: More details of the ecohydrological model (W3) is needed to show how exactly it works.

5). Page7, Line 18-19: Please move API to Materials.

6). Page8, Line 9-11:How can these two statements be justified from Fig.3d? What do R0 and Ra stand for? I assumed they represent correlations for open-loop and joint assimilation? You need to indicate it at least in the Figures.

7). Figure 5 : I suggest authors to label these sample sites on Figure 2.

8). Page8, Line 15: "marginally better than SMOS-only results", which is hard to tell from the figure.

9). Page9, Result-4.2: This section needs extra experiments using de-seasonalized data as mentioned in the major concern 2.

10). Page12, Line 26-27: There is a recent study very relevant to this statement that used GRACE-derived TWSA for Australia.

Xie, Z., Huete, A., Restrepo-Coupe, N., Ma, X., Devadas, R., Caprarelli, G., 2016. Spatial partitioning and temporal evolution of Australia's total water storage under extreme hydroclimatic impacts. Remote Sensing of Environment. 183, 43–52.

11). Page12, Line 28-29: This is likely to be attributed to 2015 El Niño impact.

---

## Author Comment (AC1) · 4 Dec 2018

This manuscript presents the results of assimilating SMOS and GRACE data, jointly and independently, into an eco-hydrological model at the global scale. Model results were evaluated using in situ soil moisture and MODIS NDVI estimates. The assimilation method seems sound and the results are reasonable and should be of great interest to researchers in the community.

[Figure]

1. My comments are mainly on how the assimilation impacts are assessed using NDVI. The authors employed MODIS NDVI to assess the impact of data assimilation on model estimates and suggested that increased correlation between soil moisture and NDVI is an indication of improved water storage estimates. This assumption may only be valid in arid and semi-arid regions; in other regions where ET is limited by available energy, changes in NDVI and water storage may not be correlated at all.

We thank the reviewer for the comments. We agree with the reviewer that vegetation mainly responds to water availability in semi-arid to arid regions. In fact, we mentioned on Page12 Line14 and Fig7, that we expected low correlation between NDVI and water storage over humid regions. To eliminate further confusion, we will mask out the humid regions in our revision.

2. I think the correlation presented in the paper may have been affected by the seasonality in the data which is easy to see in Fig.5 where increase in correlation is associated with increased seasonal amplitudes in model estimates. If the intention is to report on increased seasonality, it may be better to compute the changes in seasonality; otherwise, I would suggest the authors compute the correlation based on anomalies relative to monthly mean to eliminate the influence of seasonal variation. In either case, physical-based evidence and reasoning needs to be provided for justifying the use of NDVI in this purpose.

We agree with the reviewer that the increase in correlation is associated with seasonality in model estimates. Our intention is to demonstrate that assimilating satellite water observations help to improve model estimates of both seasonality and anomalies. As shown in Fig5b, the assimilation significantly improves the seasonality of soil moisture in eastern Brazil. On the other hand, we used NDVI as an indirect method of demonstrating improvements of soil water estimates, since in-situ measurements of soil moisture are often not representative or incomparable with gridded simulations. Since water is the controlling factor of vegetation growth in arid regions, we can expect an improved correlation with vegetation greenness after assimilation. In revising this

manuscript, we will show the improvements in anomalies to justify and further highlight the benefits of data assimilation.

3. In fact, the authors may look into the correlation between in situ root zone soil moisture and MODIS NDVI to see if and when this assumption is correct. In addition, if NDVI is truly correlated with water storage, wouldn't comparing model estimated ET with NDVI make more sense?

We understand the reviewer's suggestion, however we respectfully disagree on the need for checking the correlation between in-situ soil moisture with MODIS NDVI. The NDVI value of each grid represents the average vegetation conditions over ~25 km2, whereas the in-situ measurements only represent the soil moisture condition at point scale. As such, the average vegetation condition may have no correlation with in-situ measurements at all when, for example, comparing NDVI for a grid cell with various vegetation types to a soil moisture monitoring site located in barren patch within cell.

We applied the ensemble Kalman smoother with a one-month assimilation window to overcome the inconsistent temporal scale between observations and model. This retrospective data assimilation allows the reanalysis of state variables including all the water stores. However, the modelled ET and other variables were only affected indirectly through the water balance with updated initial conditions of water storage estimates once a month at the first day of a month. It has been demonstrated in Tian et al (2017) that there was marginal effect of soil moisture data assimilation on model ET in retrospective data assimilation. The comparison between model-simulated ET and NDVI may be considered in a future study with daily updates of ET in the model.

4. The manuscript can also be improved with more details on the model and the rationale behind pre-processing satellite data. For instance, the ecological aspect of the model is never described and no results were provided on how vegetation responded to data assimilation. And there is no discussion on why SMOS data needed to be scaled before assimilation while GRACE data were not. What are the adverse impacts of

assimilating SMOS without scaling and how are they related to any model behaviors?

We thank the reviewer for this opportunity to add more detail about data preprocessing. Our revisions will certainly clarify the reasons for rescaling of SMOS in Section 3.1.

Specifically, the transforms are necessary to achieve consistency between modelled and observed soil moisture, using the least amount of ancillary information as possible. We transform the SMOS volumetric soil moisture to relative wetness (0-1) by scaling between the driest and wettest periods observed from SMOS through time. This is because estimates of soil moisture from the W3 model are in storage depth (mm). No physical depths for soil layers are used in the model, thus model-simulated soil storage cannot be converted to volumetric soil water content. We applied the same conversion to model-simulated soil water storage to resolve the inconsistency in units between model and observations. GRACE data were not scaled because there is already a consistency in units between W3-simulated total water storage anomalies and GRACE observations.

To clarify we will add the following to Section 3.1: "Since the W3 model only specifies soil water storage in water depth (mm) rather than prescribing a physical thickness of the soil layers and porosity, the model-simulated soil water availability cannot be directly compared with SMOS soil moisture retrievals in volumetric fraction. To resolve the inconsistency between model and satellite observations in representing the near-surface soil water availability, both SMOS retrievals and W3 simulated top-layer soil water storage were converted to relative wetness (0-1) with respect to the dry and wet extremes over the 7-year period, calculated as the 2nd and 98th percentiles, respectively."

In this study, we focus on investigating the impacts of data assimilation on improving root-zone soil water storage estimates by evaluating the statistical improvements with MODIS NDVI. We did not use the NDVI or other vegetation condition estimates directly from the model after the assimilation. We will clarify in Section 3.3 as follow: "The statistical improvements in correlation between satellite-observed vegetation greenness

and soil water storage were evaluated after the joint assimilation against the open-loop results."

5. Additional comments: Page 1 Line 12: Increased correlation between vegetation greenness and soil water storage does not necessarily mean improvement on water storage estimates, unless you back this up with in situ observations.

We agree with the reviewer in general. However, we argue that the statistical improvements in correlation between vegetation greenness and soil water storage estimates can serve as an independent, indirect evaluation in water-limited regions. In particular, we demonstrate that improvements in correlation derived from the inclusion of GRACE and SMOS data in Fig 7. Such systematic improvements are unlikely to be random. Since satellite-observed NDVI represents average vegetation condition over each pixel, the in-situ soil moisture measurements at point scale cannot be directly compare with MODIS NDVI (as mentioned above in comment no. 2 and 3). This point-to-pixel comparison becomes even more problematic in the area with various land cover types. This is why we show the evaluation of soil moisture estimates with in-situ measurements as a fraction of tree cover. Therefore, we propose in this study that the correlation improvements between vegetation greenness and soil water availability in water-limited ecosystem can be considered as an independent evaluation for model performance after data assimilation.

6. Line 21: root depths do not unilaterally determine whether plants can access deeper water stores. The capillary force can also lift water up from deeper water stores to near surface soils.

We agree with the reviewer. Indeed, it is why we refer to available water storage at different layers, instead of rooting depth. Capillary rise and drainage is accounted for in the W3 model.

7. Page 2 Line 23-25: Vegetation water content only constitutes a very small part of GRACE derived TWS and thus the lag may not be related to vegetation greenness at

all.

We agree with the reviewer and we will clarify in our revisions. We mentioned that there is no time lag between GRACE observed water storage anomalies and changes in surface greenness over Australia based on the study by Yang et al. (2014). In addition, we do not mean to suggest that vegetation water content can be observed by GRACE and related to vegetation greenness. We mean that, in the water limited ecosystems, GRACE TWSA can explain changes in surface greenness both interannually and seasonally due to the ecohydrological link between water availability and vegetation vigour. In the revision we will clarify as follow:

"It has been demonstrated that GRACE-observed total water storage anomalies can explain changes in surface greenness both interannually and seasonally without time lag over Australia (Yang et al., 2014)."

8. Page 3 Section 2.1. Does groundwater interact with soil moisture?

Yes. The unconfined groundwater storage is simulated by the model considering deep drainage from soil water, capillary rise and groundwater evaporation and discharge. We will include additional description of W3 model in our revised manuscript.

9. Page 4 Line 18: a 0.25 grid or 0.5?

We thank the reviewer for the opportunity to clarify. The GRACE data were resampled to 0.25 degree grid using the JPL RL05M product. The data assimilation outputs in this study were also at 0.25 degree resolution.

10. Line 25: Is absolute value the total error of SMOS and GRACE? What is the purpose for calculating relative errors base on land cover types (i.e., Fig. 1)? Given the coarser scale of GRACE data and the fact that vegetation water content is only a small part of total water storage, I don't think GRACE errors are related to vegetation types.

We thank the reviewer for the opportunity to clarify. The relative error was calculated separately for SMOS and GRACE using their respective absolute values of uncertainty.

[Figure]

Soil moisture retrievals derived from brightness temperature from L-band emissions are affected by interference from the vegetation canopy, causing greater errors over areas with dense canopy cover. Therefore, it is important to consider these errors in the assimilation and evaluation. We agree with reviewer that GRACE error is not related to land cover types as shown in our plot (Fig1b). There is no significant difference on GRACE uncertainties among different landcover types. The main purpose is to show the relative weighting given to the model in the joint assimilation between SMOS and GRACE.

11. Page 5 Line 9-10: Using the 2nd and 98th percentiles as wilting point and field capacity can be a problem if the sites are located in very drier or wet climate where soil moisture is often restricted to one side of its full range.

We agree with reviewer that using 2nd and 98th percentiles can be problematic in continually dry or wet environments. However, we applied the same scaling to convert the model-simulated soil water storage over the same time period, which mitigates that issue. Using the statistical information to derive the wilting point and field capacity was done to avoid introducing extra uncertainties from soil property maps, which are notoriously unreliable.

12. Line 23: Were the field capacity and wilting point here derived from SMOS measurements or from the model? After the adjustment, would you convert relative wetness back to soil moisture content for assimilation?

We thank the reviewer for the opportunity to clarify. The relative wetness of SMOS, in-situ measurements and model simulation were converted using the field capacity and wilting point derived from each dataset independently. In the assimilation, we used the converted SMOS relative wetness to adjust the W3 relative wetness in the assimilation. The updated relative wetness of near-surface soil layer was converted back to soil water storage in the model representation after the adjustment and used in the initialization of model forward run for the next time step.

13. Page 6 Line 1: Adjusting temporal variance of SMOS data is not bias correction. Why did you need to do that? And why didn't you adjust the temporal variance of GRACE data to match that of W3 estimates?

We apologise for the confusion. We did not adjust the temporal variance of SMOS to the W3 model. We only converted both SMOS and W3 to relative wetness to resolve the unit inconsistency as explained above. In the revision, we will clarify as follow:

"To resolve the inconsistency between model and satellite observations in representing the near-surface soil water availability, both SMOS retrievals and W3 simulated top-layer soil water storage were converted to relative wetness (0-1) with respect to the dry and wet extremes over the 7-year period, calculated as the 2nd and 98th percentiles, respectively."

14. Line 13: equation (2). How does this update work? SMOS and GRACE have different spatial and temporal resolutions?

We thank the reviewer for the opportunity to clarify. Essentially the forecast state vector is a concatenation of all model states for all days over the given month. Equation 2 shows the state updating equations based on ensemble Kalman smoother. Since it is a retrospective data assimilation method with one-month fixed window, the model was run forward for one month for each ensemble member (driven by perturbed forcing). State updating was implemented at the end of the month using equation 2 to update all the water storage estimates for all days in a month back to the first day. The state vector xf includes all the water storage components for all days in the month (as see in Page6 Line 6). The observation vector y includes all the daily SMOS observations and one GRACE observations. The observation operator (H) includes the temporal aggregation components to convert the model estimates into observation space (Page6 Line19). The Kalman gain matrix was calculated with observation error variance and model error variance derived from the ensemble members. More technical details are explained in Tian et al. (2017). We have explained the major changes from that method, i.e. the

modification of observation operator and data source, but sought to keep explaining the assimilation method concise.

15. Line 20: why do you need field capacity and wilting point to convert soil moisture content to equivalent water heights?

We refer the reviewer to the response to item 4, 11, 12 above.

16. Page 7 Line 14: Do the two "correlation" words mean the same thing?

Yes. The "correlation" here refers to the correlation between NDVI and soil water availability derived from different dataset or models, i.e. model open-loop, joint assimilation, satellite observation only, and API.

17. Line 17: The API needs to be introduced in the data section.

We thank the reviewer for their suggestion. In revising the manuscript, we will introduce API in the data section.

18. Line 20: This sentence does not sound right. In fact, I find this whole paragraph difficult to understand.

We apologise for the lack of clarity. We hope that the following revision will make the point clearer: "The soil water availability at the integrated depth that has the maximum correlation with NDVI best explains the changes in surface greenness at each grid cell."

19. Page 8 Line 4: It looks to me that Fig. 3(b) has more points below the 1-1 line. Can you provide average r for DA and open loop, respectively? Line 9-11: joint assimilation improves SMOS soil moisture retrievals? Line 13: Fig. 4. Why are there fewer data points for the GRACE-alone plot? Line 16: This further improvement is not obvious to me. You probably should provide averaged statistics in numbers to back this up. Line 20: "less affected" is not obvious to me

We understand the reviewer's confusion here. The revised manuscript will include average error statistics to justify our statements in Line 4, 16, and 20. Our objective

here was to illustrate that the correlation of SMOS over non-forest (white circles) areas were better than W3 model open-loop. The appearance of fewer data points for the GRACE-alone plot is because that most of the sites are clustered together with not much change from open-loop after the assimilation of GRACE alone.

20. Line 23: Listing uncertainties of SMOS and GRACE in Table 1 can be misleading as they are not relative to the same set of in situ data and have nothing to do with improvements made by data assimilation. Besides, they are not referred to throughout the paper.

We thanks the reviewer for point this out, but we have referred to the results and discussed on Page 11 and Lines 1 – 5. The listed SMOS and GRACE uncertainties are for those model grid cells compared with the same set of in-situ data. The purpose of including the observation and model uncertainties was to investigate the potential reasons for the improvement and degradation, respectively.

21. Page 9 Section 4.2. I don't think increased correlation with NDVI can be counted as improvements. In many cases, it is due to increased seasonality such as in Savannas, eastern Brazil and hence increased correlation with NDVI which has strong seasonal changes. correlation in Figs. 6 and 7 should be calculated based on anomalies relative to monthly mean; otherwise, it reflects mostly the seasonality.

We agree and direct the reviewer to our response to Item 1, 2 and 3 above. We will show the improvements of anomalies in our revised manuscript.

22. Page 10 Line 8: what is exactly plant-available soil moisture storage? Root zone soil moisture?

We thank the reviewer for the opportunity to clarify. The plant-available soil moisture storage refers to the soil water availability at a depth that best explains the interannual and seasonal change of surface greenness. As shown in Figure 5, the change in water storage at different depths shows different correlation with vegetation conditions.

[Figure]

23. Section 4.3. Trend should be calculated using anomalies relative to monthly mean.

We agree with the reviewer and will calculate and include the trends based on anomalies in our revised manuscript.

24. Line 4: Can you include the correlation between soil water storage from the open loop with NDVI in Fig. 7?

We have included the difference between the open-loop and DA in Figure 6. We can include open-loop correlation again in Figure 7 if the editor considers that this would add value.

25. Page 11 Line 27. You stated earlier that "greenness can serve as a surrogate for water availability in water-limited regions". But NDVI was used to evaluate changes in water storage across the globe, regardless of climate conditions.

We agree and direct the reviewer to our response to Item 1 above. We will exclude the radiation and temperature limited regions in revising the figures and results.
* * *

---

## Author Comment (AC2) · 4 Dec 2018

The submitted manuscript by Siyuan Tian et al investigated the impacts of assimilating satellite water content retrievals on the estimation of surface and root-zone soil moisture over the globe and across different land cover types. The authors aimed at improving the accuracy of root-zone soil moisture prediction by jointly assimilating satellite-observed soil moisture from SMOS and total water storage changes from

[Figure]

GRACE into a global ecohydrological model. They then evaluated the performance of the joint assimilation by comparing against the open-loop model and alternative assimilation methods with ground-based soil moisture measurements and vegetation index.

This paper is well written, properly structured and presented, with interesting results being thoroughly interpreted by a good discussion. I believe this manuscript will be interesting to future HESS readers and contribute to the international literature. There are two major concerns that I would like the authors to address before the publication of this manuscript.

1. While GRACE-derived TWSA provides an integrated measurement of water storage changes above and underneath the earth surface, why would near-surface soil moisture derived from SMOS still be required? Don't SMOS and GRACE monitor overlap water content at near-surface? This has not been fully justified and explained in the Introduction or in the ecohydrological modelling method.

We thank the reviewer for the opportunity to clarify. The SMOS and GRACE did both include the water content at near-surface. However, the near-surface soil moisture content is highly variable both spatially and temporally. The assimilation of monthly GRACE data alone has little impact on the estimation of near-surface soil moisture (Zaitchik et al., 2008, Tangdamrongsub et al., 2018; Li et al., 2012; Girotto et al., 2017). The assimilation of daily SMOS observation with higher spatial resolution together with monthly coarse GRACE data can better disaggregate the vertical distribution of water storage into different components. As demonstrated in Tian et al., (2017), the joint assimilation of SMOS and GRACE provides constraints on both the total water storage estimates and surface soil moisture estimate, as a result providing more accurate root-zone soil moisture and groundwater storage estimates.

In our revised manuscript, we will include the following justification in the introduction: "Conversely, GRACE-observed total water storage anomalies were successfully assimilated or otherwise combined with model simulations for improved deep soil and

groundwater estimation (Zaitchik et al., 2008; Khaki et al., 2017; Schumacher et al., 2018; Tangdamrongsub et al., 2015; Girotto et al., 2017; van Dijk et al., 2014; Tangdamrongsub et al., 2018), but with typically marginal improvements for surface and shallow soil moisture (Tangdamrongsub et al., 2018; Li et al., 2012; Girotto et al., 2017; Tian et al., 2017). This is due to the highly variable nature of near-surface and shallow soil moisture in space and time, which has little influence on the GRACE signal. Recently, near-surface soil moisture and total water storage observations were jointly assimilated into a water balance model over Australia and demonstrated consistently improved water storage profile estimates, especially in the root-zone soil moisture estimates (Tian et al., 2017). The use of satellite-observed daily near-surface soil moisture has been demonstrated to better disaggregate shallow soil moisture and groundwater change from GRACE-observed total water storage change because of the different temporal dynamics."

2. Following up Reviewer#1's major comment on assessing assimilated soil moisture using NDVI, I do agree Reviewer#1 that extra experiments of correlation analyses based on de-seasonalized times series of all data are required. Although I agree with the authors that the improvements of the modelled root-zone soil moisture over only ET limited regions are likely due to increased seasonality, authors may need to show how the methods proposed in this study could improve root-zone soil moisture in the long run without the effect of seasonality.

We agree. We will show the improvements on anomalies in revising the manuscript to address the concerns from both reviewers.

My specific comments are as follows: 1). Page1, Line 4:Do you have references to confirm this? Some people believe GRACE-derived TWSA is mainly dominated by soil moisture variation over many places.

We agree with the reviewer that GRACE-derived TWSA can be dominated with snow and/or soil moisture at different locations. However, there are studies that show that the

changes in GRACE TWSA mainly come from changes in groundwater such as Rodell, Velicogna and Famiglietti (2009), Famiglietti et al. (2011) and Voss et al. (2013). To be more precise in the abstract, we will modify this as follow:

"In contrast, GRACE (Gravity Recovery and Climate Experiment) mission detected the variability in storage within the total water column, with no vertical resolution. Root-zone soil moisture, often the main interest in agriculture and ecology, cannot be separated from GRACE observed total water storage anomalies without ancillary information on surface soil moisture or groundwater changes."

2). Page3, Line 9-18: Introduction is well presented, however, this paragraph of objectives could be improved by clearly numbering each objective such as 1). . .. 2). . ..3). . .. This will make it easier for future readers to get straight to the points.

We thank reviewer for the suggestion and we will clearly define the objectives of this study at the end of Introduction section in revision:

3). Page3, Line 27: includes aÌĆA Ì̱TËĞ> including, and these .

We are not entirely sure what was originally shown in this comment, but suspect the suggestion was to modify this sentence into:

"The $0.5 \times 0.5$ WFDEI (WATCH Forcing Data methodology applied to ERA-Interim) meteorological forcing data set (Weedon et al., 2014) used in this study including radiation, air temperature, wind speed, and surface pressure, and these were resampled to be consistent with the resolution of precipitation at 0.25."

4). Page3, Line 21-30: More details of the ecohydrological model (W3) is needed to show how exactly it works.

We agree. While the W3 model has been described in numerous other studies, we will include in our revisions the following additional explanation in revision at Section 2.1 Line 28:

"Precipitation is assumed to be the only water input into the system. The precipitation enters the grid cell through the vegetation and soil moisture stores and exits the grid cell through evapotranspiration, run-off or groundwater discharge. Each grid cell contains a mix of land cover classes (Hydrological Response Units; HRUs) and is conceptualized as a catchment that does not laterally exchange water with neighbouring cells. Different vegetation has different degrees of access to soil water. Soil and vegetation water and energy fluxes were simulated separately for deep-rooted and shallow-rooted vegetation to consider different rooting and water uptake behaviour. The soil water store was partitioned into three layers, namely, top, shallow and deep soil to describe the plant available water, approximately 0–5cm, 0.05–1m, and 1–10m in depth respectively. The unconfined groundwater and surface water stores were simulated comprising the evaporation, discharge and runoff at grid cell level."

5). Page7, Line 18-19: Please move API to Materials.

Agreed. We will move API to the data section in revision.

6). Page8, Line 9-11:How can these two statements be justified from Fig.3d? What do R0 and Ra stand for? I assumed they represent correlations for open-loop and joint assimilation? You need to indicate it at least in the Figures.

Thank you. Yes, the Ro and Ra are the correlation for open-loop and joint assimilation, respectively. We will improve the figure caption to specify this.

7). Figure 5 : I suggest authors to label these sample sites on Figure 2.

Thank you. We take the reviewer's suggestion and we will label these sites on Figure 2.

8). Page8, Line 15: "marginally better than SMOS-only results", which is hard to tell from the figure.

Agreed. We will include the averaged statistics in the revision.

9). Page9, Result-4.2: This section needs extra experiments using de-seasonalized data as mentioned in the major concern 2.

Agreed. We understand reviewer's concern and we will include the de-seasonalized experiment in the revision.

10). Page12, Line 26-27: There is a recent study very relevant to this statement that used GRACE-derived TWSA for Australia. Xie, Z., Huete, A., Restrepo-Coupe, N., Ma, X., Devadas, R., Caprarelli, G., 2016. Spatial partitioning and temporal evolution of Australia's total water storage under extreme hydroclimatic impacts. Remote Sensing of Environment. 183, 43–52.

We thank reviewer for this reference and we will include the citation in the manuscript as follow: "After a sharp recovery from the Millennium drought with an extremely wet period from 2010 to 2011 (Leblanc et al., 2009; van Dijk et al., 2013a, Xie et al., 2016), drought returned to eastern Australia with a decrease in soil water over 15 mm/yr estimated from both model open-loop and joint assimilation (Fig. 8a and 8b)."

11). Page12, Line 28-29: This is likely to be attributed to 2015 El NinÌČo impact.

We thank reviewer for the suggestion. We will include the following statement in the manuscript as:

"A decline in NDVI of more than 0.025 units per year was observed for the majority of middle and eastern Australia due to the developing soil water deficit (Fig. 8d), which is likely due to the widespread rainfall deficits caused by the El NinÌČo 2014-16 and further amplified by the Indian Ocean Dipole 2015."